# What should pulmonary rehabilitation look like for people living with post-tuberculosis lung disease in the Bishkek and Chui region of the Kyrgyz Republic? A qualitative exploration

Maamed Mademilov [1,2] Gulzada Mirzalieva,[1,2] Zainab K Yusuf [3,4]
Mark W Orme [3,4] Claire Bourne,[4] Azamat Akylbekov [1,2] Amy V Jones [3,4]
Ruhme B Miah,[3,4] Rupert Jones,[5] Andy Barton,[3] Dominic Malcolm,[6]
Talant Sooronbaev,[1,2] Sally J Singh[3,4]

MM and GM contributed equally.

For numbered affiliations see end of article.

**Correspondence to**
Dr Maamed Mademilov;
mademilov@gmail.com

## ABSTRACT

**Objective** After experiencing tuberculosis (TB), many people develop post-tuberculosis lung disease (PTBLD). Pulmonary rehabilitation (PR) centrally comprising of education and exercise is recommended internationally for people living with chronic respiratory diseases. However, no such service exists in Kyrgyzstan. This study investigated the opinions of healthcare professionals who would be expected to be potential future referrers to PR and adults living with PTBLD about what a PR programme could look like in Kyrgyzstan.

**Design** A qualitative study using interviews and focus groups. Grounded theory and thematic analysis were used for data collection and analysis.

**Participants** 63 participants; 15 referrers (12 male, 3 female; 12 pulmonolgists, 3 TB specialists) and 48 adults (26 male, 22 female) living with PTBLD.

**Setting** Participants were recruited from hospital settings in Bishkek and Chuy Region, Kygryzstan.

**Methods** Fifteen semistructured interviews were conducted with referrers and nine focus group discussions were conducted with adults living with PTBLD.

**Results** Five key themes were developed: (1) living with PTBLD; (2) attitude to PR, which emphasised the perceived importance and potential benefits of implemention; (3) barriers/facilitators to PR, which included time and cost, and the importance of appropriate communication in enabling participation; (4) interventional components of PR, which described culturally and demographically appropriate physical activities including rhythmic movements, dance and volleyball; and (5) psychosocial support, which demonstrated the importance of psychological support for patients coping with the effects of stigma.

**Conclusions** Potential referrers and adults living with PTBLD expressed their support for the implementation of PR. The culture-specific and population-specific issues highlighted in this work demonstrate the need to address stigma and provide certain types of exercise training/education modules for this specific clinical population. In other respects the currently known attitudes/barriers to

### Strengths and limitations of this study

► The opinions of patients with post-tuberculosis lung disease (PTBLD) and healthcare professionals (tuberculosis specialists and pulmonologists) were deliberately taken into account to develop a pulmonary rehabilitation (PR) programme.
► Participants were diverse in terms of demographic characteristics, severity of the disease, experience and social status.
► The study was conducted in one geographical region in Kyrgyzstan, therefore, the implications reported need to be tested to show applicability to other cultural contexts.
► The study focused on PR for PTBLD rather than the broader spectrum of lung diseases.

PR, identified in Western research, appear to apply. The principles of culturally adapting PR may be helpful for those looking to establish similar clinical services in other low-income and middle-income countries and in Central Asia in particular.

**Trial registration number** ISRCTN11122503.

## INTRODUCTION

Worldwide, 10 million people suffer with tuberculosis (TB) and 1.5 million people die from TB annually. The burden of TB on individuals and society falls predominantly in low-income and middle-income countries (LMICs)[1] where weak economic conditions aggravate the situation of people living with the disease, leading to disability and early mortality.[2 3] Kyrgyzstan is an LMIC in Central Asia with a high poverty level (20.1%).[4 5]

According to WHO, the relative incidence rate of TB in Kyrgyzstan in 2019 was 144 per 100 000, the highest rate in the WHO European Region,[6] as well as having one of the highest respiratory mortality rates.[7]

Many adults who have experienced TB go on to develop post-tuberculosis lung disease (PTBLD). While chronic respiratory diseases (CRDs) are common in the Kyrgyzstan, PTBLD is the most challenging lung condition to address, because of public fears about the risk of infection to others, contagion which in turn leads to stigmatisation, as evidenced by the issuing of a certificate of full recovery for these patients. People with PTBLD experience decreased exercise tolerance and increased cough, shortness of breath and general weakness. This is often accompanied by feelings of social isolation and depression.[8 9] There is evidence of persistent stigmatisation of people living with TB even after recovery[10 11]; compounding problems such as social exclusion, unemployment, unstable housing and limited access to medical care.[10 12 13] Depression, anxiety and somatic symptom disorders have a particularly negative impact on the standard of living of people in LMIC such as Kyrgyzstan.[3 14 15]

Kyrgyzstan declared its independence from the Union of Soviet Socialist Republics (USSR) when it collapsed in 1991. The Soviet model of healthcare, which was centralised and provided universal free access to medical care, was designed for the prevention and early warning of diseases. During the Soviet era, people living with PTBLD across the USSR attended Pulmonary rehabilitation (PR), where services ranged from 3 weeks to 3 months and included breathing exercises, prescribed walking, massage and various ball games. In addition, there were other alternative treatment institutions such as sanatoriums, salt mines and alpine resorts.

After the collapse of the Soviet Union, healthcare systems in independent states have experienced an economic crisis, an outflow of specialists, a decrease in the quality of medical care, a shortage of drugs, and an increase in the number of diseases such as TB and HIV. While the rehabilitation centres opened during the Soviet era continued to operate after the independence of Kyrgyzstan, due to a lack of funding, the quality and usefulness of the services provided has reduced.[16] Rehabilitation is rarely available to patients. Consequently, the burden of TB and PTBLD has continued to grow since Soviet times.[17] At present, the treatment process for these individuals is exclusively pharmacological (lasting between 6 months and 2 years) and, with no post-treatment care pathway, they are often left in a state of limbo whereby they are no longer infectious but are living with chronic symptoms and associated stigma. It is important to note that PR is not implemented anywhere in Central Asia.

The importance of PR is well evidenced for people living with CRDs, including for adults living with PTBLD.[10 18 19] PR involves initial assessment followed by individualised therapy (exercise training and education) to improve physical and psychological well-being, and develop a long-term commitment to healthy behaviour.[20 21] PR improves shortness of breath, exercise tolerance and quality of life and reduces the number and duration of hospitalisations linked to exacerbations.[18 22 23]

However, the introduction of PR in LMIC needs to be sensitive to both the local populations receiving care and the local context. Access to PR is typically through a referral process whereby healthcare professionals (HCPs) introduce PR to patients. In many LMIC, including Kyrgyzstan, pulmonologists and TB Specialists work directly with adults living with PTBLD and will likely be involved in referring patients for PR. Consequently, this study combines the views of potential PR referrers and adults living with PTBLD in Kyrgyzstan to identify both the potential acceptability and most appropriate design for PR.

### Aims

To explore the views and experiences of potential referrers and adults living with PTBLD in Kyrgyzstan to inform the development of a culturally appropriate PR service.

## METHODS

### Patient and public involvement

As respiratory clinicians, we drew on our experiences of talking to people living with PTBLD when designing the study, in particular the interview schedule and logistics. Interviews and focus groups (FGs) were organised based on work schedules and family commitments. This work was conducted to provide more in-depth insight into the views of patients to improve the quality of care they receive, such as the provision of appropriate PTBLD-specific PR. Patients were not directly involved in formulating the interview schedules or research question in the traditional sense of patient and public involvement. It was through informal conversations with patients and clinical consultations with patients that the importance of directly involving patients in the design of their PR arose; not least because of the stigma and alienation experienced as a result of their PTBLD. Patients who were involved in the research and resulting PR trial will be invited to support future dissemination activities. By the nature of our qualitative research, the findings of this study are being used to develop a patient-informed PR programme in accordance with the principles of PPI. The findings of this study will be disseminated to participants and the wider public through awareness-raising activities in TB hospitals to help boost the profile of PR and inform the wider public about PTBLD-specific rehabilitation. The findings will also be disseminated through national and international conferences. The results of this study are expected to inform the Kyrgyz healthcare system by introducing a culturally appropriate rehabilitation service informed by patients themselves.

### Participants and data collection

In line with the research protocol, HCPs in regular contact with adults living with PTBLD were invited to

participate and were recruited from five sites in the Chuy region that offer PTBLD treatment (National Hospital, National TB Hospital, City TB Hospital, Chuy region TB Hospital and Alamedin District TB Hospital). A total of 17 HCPs were invited to take part. Two HCPs declined to participate due to lack of time, leaving 15 individuals who participated in semi-structured interviews (12 females, 3 males; 3 pulmonologists and 12 TB specialists).

Participants for FG discussions were invited from eight sites (National Hospital, National TB Hospital, National Centre of Cardiology and Internal Medicine (NCCIM), City TB Hospital, Chuy region TB Hospital, Sokuluk District TB Hospital, Alamedin District TB Hospital and Kant District TB Hospital). All participants had previously received and finished basic treatment fo PTBLD. In total, nine FGs were organised. Eighty people were invited to participate, of whom 48 took part (26 men and 22 women) in groups of between 4 and 8. Participants came from a range of socio-economic backgrounds (including some homeless participants) and geographical locations (urban and rural).

The sampling strategy was a combination of targeted and convenience sampling. We targeted only medical specialists whose clinical practice directly involved dealing with PTBLD patients. All participants provided written informed consent. The interviews with referrers were conducted face-to-face at the participants' workplace in various healthcare facilities, both in urban (national centre, city centre) and rural settings between October 2019 and March 2020. The FGs were conducted on the premises of the department of respiratory medicine of the NCCIM between October 2019 and January 2020. During the FGs, participants were free to interact with each other, offer peer-to-peer support, ask and answer questions, and this method enabled patients to provide extended discussions of topics that were of particular interest to them, such as the problems around stigma. The interviewer (MM) was a pulmonologist in Bishkek, Kyrgyzstan with previous experience in qualitative research. Interviewer MM acted as facilitator during FG. MM did not have any relationship with participants prior to study commencement and equally, participants did not know MM prior to the interviews.

The interview guides were structured using a narrative approach with additional prompts. Different interview guides were used for interviews and FG (see interview guides in online supplemental appendixs 1 and 2). To improve and adapt questions and prompts, notes were taken and discussions of transcribed data were regularly held with the research team. The interviews and FGs were conducted in Russian and Kyrgyz languages according to participant preference. The researchers used audio-recordings to collect the data. The average interview duration was 30 min, and for FGs 50 min. There were no follow-up interviews or FGs.

The the interview recordings were transcribed verbatim and then translated into English to enable the cross national team to fully participate in the analysis. The

translation was carried out by a Kyrgyz team who were able to understand and interpret all expressions, and conveyed such details as laughter, expressions, etc in the transcripts. Specialist and local terms were converted into an accessible form for understanding in English. Each interview and FG transcript was assigned a sequence number. Transcripts were anonymised and were not returned to participants for verification.

## Data analysis

The theoretical framework used was grounded theory and the data were analysed using thematic analysis.[24] The flexibility and explorative character of thematic analysis makes it particularly appropriate for comparing and contrasting qualitative data drawn from different sample groups, and where researchers have variable levels of understanding of the life-worlds of different participant groups.

Each step of the analysis was carried out independently by the Kyrgyz and UK teams, followed by joint discussions, and agreement on the key themes. The first step involved reading and rereading the transcripts several times (in their original or translated form), and writing a brief summary of the data. Step two involved manually creating codes (software packages were not used). There were seven coders in total; three in Kyrgyzstan and four in the UK. The combination of similar phrases and sentences were highlighted in the same colour, placed under the same code and a description of its contents was developed. A total of 21 initial codes were created. The third step involved refining the themes by grouping together similar codes. The fourth step involved collating all of the relevant quotes and clarifying how each data extract related to the different themes. The fifth step involved developing a detailed description of themes and subthemes together with the use of relevant quotes.

Summary findings and interpretations encapsulating the perspectives of both referrers and adults living with PTBLD are presented. For the purposes of this article, data have been organised into five overarching themes, impacting a range of different PR-related issues. Quotes were selected from frequently encountered codes during the interviews and FGs and chosen as the best representation of the analytic point. Recruitment targets were determined a priori according to the norms of the respective methods. Partly due to the delays imposed by transcription, translation and cross-national analysis, but also due to the relative ease of access, we recruited up to these targets. Data saturation was achieved, but known only retrospectively as the data analysis progressed.

## RESULTS

Referrers and respondents living with PTBLD were positive about the potential introduction of PR. They identified a range of barriers/facilitators that would need to be considered in the design and implementation of any such programme. A range of potential PR components were

identified, as well as the need to tailor PR to individual physical conditions, provide effective communication and education, and be sensitive to the patients' psychosocial state. It was also noted the importance of including psychological support in the PR, which will be specific to patients with PTBLD who have stigma and psychological problems. Respondents living with PTBLD shared their illness experiences, such as symptom burden, challenges faced and patterns of exercise and physical activity, and provided a vision of how a PR programme should best be designed. Themes obtained in the thematic analysis are presented in the diagram (see diagram in online supplemental appendix 3).

### Living with PTBLD

Respondents living with PTBLD identified a range of expected symptoms including, 'weakness and shortness of breath' (#FG2). However, the severity of symptoms varies for people living with PTBLD and affects their daily life in different ways. For some, symptoms do not present any difficulties for day-to-day living; 'At the moment I am walking freely, doing household chores' (#FG5). But how strongly the respiratory symptoms of PTBLD affect quality of life was noted by another respondent: "…with any physical movement I have shortness of breath… shortness of breath is strong" (#FG9).

Many respondents living with PTBLD worry about contracting TB again. They strictly monitor their symptoms and when their condition worsens or new symptoms emerge, they turn to medical staff for advice:

> …I do everything that the doctor said, every year I go to the clinic to [the] TB specialist, cough, sneeze, get a yearly consultation, every year I go to X-rays (#FG1)

In addition to recognising the efficacy of traditional treatments, some respondents resort to alternative treatments; '…I drink rose hips, sometimes with lemon, I eat a spoonful of honey all the time, I try to drink milk, yogurt' (#FG1). Many respondents choose spa treatment: '… I went to the salt room, in the sanatorium, Kyrgyz seaside' (#FG1). Others noted that movement and physical exercise in general improved their well-being:

> I…try to walk, sometimes I walk from morning to evening. I believe that walking, in general, physical activity is a huge benefit (#FG5)

Particularly in rural communities, where agricultural work is common, day-to-day living is relatively active. Others noted how activity was embedded into their lifestyle—'I walk a lot, do not use the washing machine, cook and wash' (#FG5)—or did certain activities at home: '…I do breathing exercises so as not to cough. In these moments, gymnastics helps me' (#FG2).

Respondents have become accustomed to living with respiratory symptoms; 'I wanted to live and breathe deeply, we have already forgotten what it is, but we want it so much' (#FG1). They are frequently proactive, seeking to improve their condition; '…we make different attempts,

we want to get better. We need life' (#FG2). Consequently there is a significant demand for the implementation of a service like PR, which can help improve health-related quality of living.

### Attitude to PR

All referrers noted the importance of implementing PR and recognised that the current absence of services meant that, post-treatment, people living with PTBLD were left without effective guidance:

> Rehabilitation is the process necessary after completion of treatment for complete recovery (#R12)

This sense of importance was confirmed by the FG data, whereby participants noted the potential of PR for improving their quality of life, increasing exercise tolerance and reducing the severity of symptoms: 'If I constantly move, I feel good, nothing hurts' (#FG8). However, some have a limited awareness about which types of activities are suitable—'…you can or cannot play sports…, I don't know…' (#FG4)—or how to participate in these activities safely; '…we don't know what movements to do and how to do them correctly' (#FG2).

Consequently, FG respondents noted the importance of receiving trustworthy, expert advice about PR and qualified assistance from medical personnel; 'health professionals need to be trained so that I can come and ask, and not receive the answers that I know' (#FG1). HCPs should also adapt PR to individual needs: 'The main thing is that the coach is good, that he knows how to give the right load' (#FG5).

Referrers noted the value of helping people to improve physical function and adapt to social life. As one referrer noted, PR would:

> [lead to] an increase in tolerance to physical exertion, it would be like an addition to the main treatment… then it would reduce the number of complications, the number of exacerbations in general… so that the diseases do not become more complicated, do not worsen, as it seems to me that PR has a huge role [to play]. (#R9)

Respondents living with PTBLD generally exhibited a positive attitude towards PR with many expressing an interest in participating: 'I will gladly come' (#FG1). Another respondent stated: 'Yes, absolutely necessary' (#FG3). The FGs also revealed that some people might not want to take part in PR; for example, people who believe they are relatively healthy or have mild symptoms may be uninterested and see no benefit in participating.

### Barriers/facilitators to PR

HCPs identified relatively few barriers to referring patients to PR. Only one fundamental issue was identified as precluding PR participation; namely the presence of concomitant diseases (#R11) or unstable conditions (#R9).

Referrers did, however, note that some personality types—'phlegmatic people' (#R8), those whose behaviour was erratic, 'forgetful, shy' (#R11)—were not well suited to PR and that this would affect uptake. The importance of personality type was also noted by FG respondents:

> Among the patients there are aggressive patients who run into others for no reason, and there are calm ones on the contrary, who prefer more privacy (#FG7)

A further consideration was the social situation of some adults. It was noted that many migrate (primarily for economic reasons) after finishing treatment. It was also noted that, 'among the patients with TB, we have former convicts and drug addicts' (#R10), 'alcoholics, homeless people' (#R4) and so PR opportunities would not be taken up by everyone.

More generally, cost and time were identified as major considerations. A number of referrers argued that PR must be free to encourage attendance and meet the expectations of those living with PTBLD (#R5, 11, 13). FG respondents felt the same (#FG1) and emphasised the potential financial burden of implementing PR: 'I wish there was a discount. Many patients do not work, there is little money…'(#FG2).

Personality types and the social situation were identified as interdependent factors. One referrer noted that attitudes from those with PTBLD were affected by the experience of having TB; '…because our patients have been treated for a long time, they are asocial for a very long time' (#R1). Others felt that the prolonged treatment period often experienced would make people reluctant to volunteer for further engagement. Some referrers noted that adherence could be affected by slow positive effects on health, which can lead to a loss of interest:

> …they don't feel [the effects] right away… gradually if this is repeated regularly, you will feel a positive effect (#R2)

The two primary solutions proposed for these issues were educating the patient with a 'conversation about the need for post-treatment rehabilitation' (#R1), and attending to the 'dynamics' and 'subjective feelings' (#R2) generated in the PR environment that will help strengthen adherence to PR.

Referrers talked about the need for effective doctor-patient communication for the success of PR. In the context of referring a patient to PR, interviewees noted that the authority and recommendations of the attending physician will have a very important impact on a person's decision to participate in PR:

> If you explain normally to a person, any person will do what the doctor recommends, because in our society, as a doctor, as long as we have authority, patients believe us. (#R3)

Most respondents noted that they would follow recommendations to improve their health:

> You need to do whatever your doctors tell you, take the right medication, eat right (#FG3).

However, positive clinician–patient relations cannot be taken for granted as some of the FG respondents recalled unfavourable experiences with medical staff in the past. Being open and friendly was seen as an important source of motivation for those living with PTBLD;

> …if you communicate openly with them, they are also open patients, and they will be happy to hear from you… happy to know any information. (#R2)

However, this alone would not be sufficient to mitigate experiences of stigma, which created a closed-minded attitude among those affected. Referrers explained how this could affect interactions with other HCPs:

> …because even if we write a certificate for them, … [that] they are not contagious to others, sometimes in the consultation department or even in the hospital they look at patients differently and say 'you are ill with TB' (#R12)

### Interventional components of PR

While all referrers and FG respondents noted the importance of implementing physical exercises, a broad range of specific exercises were described. Referrers noted that walking was 'the simplest and cheapest method' (#R2), while swimming and Nordic walking were commended for activating a wide range of muscles (#R9, #R15). Other activities included rhythmic movements (dancing, wushu [Chinese Kungfu]), ball sports (volleyball, basketball) and recreational activities (running, cycling, gymnastics). Like many countries, Kyrgyz culture is heavily influenced by dance, and the national dance 'Kara Jorgo' ('Black Stallion') was cited as an activity that may be suitable to include in PR (#R12). Referrers argued for '…almost all the exercises that involve the respiratory muscles, [and] the muscles of the shoulder girdle… especially the muscles of the arms and shoulders' (#R9). They further noted the '…need to try everything…' (#R5) because different people like different forms of exercise.

Referrers and FG respondents also supported singing as part of the rehabilitation, though there were several opposing opinions. For instance, FG participants argued that 'singing songs can be added to rehabilitation… it's good for the lungs' (#FG5), while referrers noted that attention should be paid to contraindications and severe concomitant diseases, which can lead to serious consequences such as internal bleeding:

> If they have changes in the lungs, if they have caverns, then… singing can somehow affect the deterioration of their condition (#R1)

But of greater concern to the referrers and FG respondents was that the intensity of activity should be suitable to the individual. Exercises needed to be done, '…not at such a pace, but … sparing, light' (#R4). As one FG respondent remarked, '…don't overwork' (#FG4).

Referrers rejected '…the dances where they jump intensively there, run there,' (#R9) and recommended activities such as:

> …rhythmic dances, then slow… they can still use the respiratory muscles in their work and [this] may, it seems to me, be useful for recovery (#R9)

Additionally, FG respondents emphasised the importance of developing the correct schedule and suggested having both morning and evening classes because, 'sometimes they [people] are not allowed to leave work' (#FG1). Others noted that the time pressures experienced by those who work or have domestic or childcare duties meant that it would be good if 'some exercises can be done at home' (#R11).

Finally, referrers stated that where education was offered to those with PTBLD, verbal delivery was preferable to reading or text:

> …video lectures should be organized in all clinical specialized pulmonary units. They should be concise, lasting 10–15 min (#R6)

Despite various challenges, the strength of desire to overcome the consequences of PTBLD was evident in the respondents' willingness to spend their limited time and money to participate in PR (#FG1). The benefits of PR were highlighted: 'the time spent on recovery is not in vain. To do this, you can postpone other activities' (#FG5).

### Psychosocial support

All TB specialists noted the need to address psychological problems (eg, depression, becoming withdrawn or aggressive), personal and relational issues (eg, with family) and to help people living with PTBLD adapt to social life. According to referrers, people may experience stigmatisation from a variety of sources. Some people 'leave the family' (#R13) due to the stigma of illness, while others are afraid that they will meet people they know at the hospital; '…[patients] tell us where and in which hospitals their relatives, friends work and do not want to visit [these places]' (#R15). Another noted that, 'stigma discrimination is very common among our patients, and I think that psychosocial support can be connected to PR' (#R2).

This was supported by the FG data, where people shared their experiences of PTBLD-related stigma. Several participants said they experienced difficulties finding a job; 'I tried so many times to get a job, they didn't take me' (#FG7). Some people struggle to recover and after treatment and discharge, they are subjected to stereotyping, affecting their relationships with their loved ones. For example,

> My neighbor…told her (daughter-in-law) and the children not to come to me, that I was contagious, and to other neighbors that I needed to be evicted. (#FG3)

Indicatively, it was suggested PR should be done 'anonymously so that no one knows' (#FG1).

Consequently, both referrers and FG participants placed significant emphasis on the role of psychological support within PR interventions. It was argued that some psychological support would be provided through the social nature of some PR activities. Where rehabilitation takes place in a group, all adults with PTBLD 'will be equal … to each other, they will talk, they will communicate' (#R3). Conducting group PR discussions could provide mutual support because, '…until a person gets sick, they do not understand others, and there everyone understands each other, they talk, they share secrets' (#R11). There was also a call for more formal, professional psychological support; 'we need psychological therapy' (#FG7).

Thus, it was not only argued that PR needed to incorporate psychological support but that it offers a good opportunity to provide the kind of support that was more generally needed by people living with PTBLD. As one FG respondent noted, 'The main thing is psychological support. Sport is sport, but warm words cannot be replaced by anything' (#FG5).

### DISCUSSION

The need to develop appropriate PR services for people living with PTBLD, and who reside in LMIC, is an important issue that has scarcely been studied. Kyrgyzstan is leading the development of PR among many other post-Soviet countries and in Central Asia, where there is currently an absence of rehabilitation services. Consequently, about how to successfully implement PR may have significant implications for other LMIC or low-resource contexts. This qualitative study synthesised the views of adults living with PTBLD, and the TB specialists and pulmonologists who would initiate referral, to explore how PR could be tailored to maximise uptake in the context of Kyrgyzstan. The results indicate both a general need and willingness to embrace PR, as well as key barriers (eg, cost, time) and facilitators (eg, staffing, scheduling and delivery mode) to uptake, and specific guidance on the components of intervention, including the role of psychosocial support. PR was viewed as playing an important role in improving both people's physical and mental health.

The integration of PR into the treatment pathway should be supported by strong long-term relationships between patients and clinicians, primarily a TB specialist or pulmonologist. The data suggest that a general inclination to respect the authority of medical professionals, and a widespread compliance with treatment recommendations, will be enhanced by an open and honest attitude towards patients. Good communication by HCPs is needed to build rapport and trust with a cohort who may have experienced discrimination. There is a need to include PR in doctor-patient conversations early in the TB treatment process so that those who go on to develop PTBLD are aware that PR is a necessary and valuable part of their overall recovery and reintegration into society.

Because of the contrasting medical guidance at the treatment and rehabilitation phases for TB (ie, initially rest and latterly activity), clinicians should be transparent and consistent in promoting awareness of the benefits of PR early on during treatment. This will likely boost the uptake to PR.

In particular, continued patient engagement will be influenced by social factors. Because social attitudes prevent some patients from participating in hospital-based PR services, it is vital to design a recruitment pathway and appointment logistics that are sensitive to issues relating to stigmatisation. For example, it may be suitable to conduct PR classes at quieter times of the day, to help patients arrange transport or to offer overnight accommodation for those travelling far from rural regions. It would be desirable to provide advice about exercise that can be done at home and particularly about what level of exertion is both beneficial and safe. Many people living with PTBLD experience financial difficulties,[25] and either struggle to find employment or secure only relatively low-paid jobs. Therefore, when organising PR for such a population, discounted or low-cost provision is essential. It is significant that participants preferred modes of exercise training that do not require expensive equipment and could be undertaken in home or community settings, such as walking, dance and balls sports as this facilitates the implementation of PR in LMIC.

Educational components of PR should be tailored to the needs of the particular audience. In parts of the world where PR is part of standard care, education has traditionally been delivered in a lecture style but the data suggest that information provided verbally, with supporting images, and in 'bite size' amounts, would be more suitable for the PTBLD population in Kyrgyzstan. This fits with an ongoing transformation of PR education delivery in other parts of the world, such as in the UK, to move away from didactic deliveries, with the educational and cultural backgrounds of PR attendees considered.[26]

Many previous studies have focused on stigma related to the infectious nature of TB and the lack of awareness within the general population that these individuals are no longer contagious after completing their drug treatment.[11] Studies conducted in LMIC such as Vietnam[27] and India,[11] reported that TB is often deemed a 'death penalty' and perceived as a 'dirty disease'. Studies in LMIC have shown that the effects of stigma can both influence a patient's treatment and care-seeking behaviour[28 29] and undermine patients' trust in HCPs.[30] A previous Kyrgyzstan study found that TB patients frequently experience discrimination from HCPs who sought to avoid interaction.[15]

A major lesson from this study is that individuals with PTBLD may continue to experience discrimination, including from HCPs who deny them advice and make it difficult for them to visit healthcare facilities. PR offers an opportunity for these individuals to receive care from HCPs free from prejudice. Incorporating educational sessions will help them understand their disease and the social attitudes they experience. In this regard, both sets of respondents emphasised the importance of incorporating psychosocial support as part of PR to help people cope with the effects of stigma. Incorporating group sessions in a safe, non-judgemental environment offers peer-peer support and will enable patients to share their experiences openly.

Another way to promote PR attendance and adherence is to embed culturally and demographically appropriate activities. The core PR components of aerobic exercise typically include walking/cycling, strength training and education.[18] The results of this study suggest that the generic principles of PR can be adapted to the local context or population. For example, respondents spoke about the popularity of ball sports such as volleyball, reminiscent of the rehabilitation offered in the Soviet era. Conventional exercises like squats can be 'gamified' into a volleyball dig action to increase enjoyment and motivation. Adaptations are likely to be most successful where they also help patients' understanding of how these exercises can translate into benefits in daily life for example, taking part in social activities or preparing for employment.

Respondents supported the incorporation of rhythmic movements such as dance performed at an appropriate intensity. A national dance such as Kara Jorgo, can both be scaled to each person's physical condition and offer a culturally relevant and enjoyable form of exercise to strengthen muscles involved in the mechanics of breathing. Dance has been shown to improve health, including physical performance and mood.[31 32] Taking part in socially conventional exercises will challenge stigma and show people living with PTBLD how PR is also relevant to everyday living.

In addition to dancing, singing is also an important part of the Kyrgyz culture. Unlike dancing, singing had a mixed reception from participants some of whom were enthusiastic and some of whom said it would not be popular.[33] Although singing therapy has been tested previously, with evidence suggesting that it can improve well-being for adults with Chronic Obstructive Pulmonary Disease (COPD),[34 35] evidence relating to the effects of singing for PTBLD populations is lacking. Consequently, further research is required before implementing singing as part of PR in the contexts of PTBLD and Central Asia.

## Limitations

The study was conducted in one geographical region in Kyrgyzstan, therefore the applicability of the implications reported require empirical verification. The absence of holistic treatment for this group of patients and the problems of care do not provide a complete picture to understand their demographic characteristics and severity of the disease, and thus it is unclear if the study results can be generalised to other people with PTBLD in Kyrgyzstan. The study focused on PR for PTBLD rather than the broader spectrum of lung diseases, but it is important to note that there is no evidence to suggest that the implementation of PR for PTBLD is likely to be easier in relation to this group, due to the stigma associated with the condition.

## CONCLUSION

PR is embedded in routine clinical care in Western countries with unequivocal evidence of its benefits. In this qualitative exploration, potential PR referrers (TB specialists and pulmonologists), as well as adults living with PTBLD, were overwhelmingly positive towards the implementation of PR in Kyrgyzstan. The culture-specific and population-specific issues highlighted in this work demonstrate the need to address stigma and provide certain types of exercise training/education modules for this specific clinical population. In other respects, the currently known attitudes/barriers to PR, identified in Western research, appear to apply. The act of culturally adapting PR may be beneficial for other settings looking to establish PR as a clinical service or evaluate PR in clinical trials in low-resource settings. The results of this study can be applied to provide a basis for the implementation of PR programmes and to evaluate the effectiveness of such programmes for people with PTBLD. Future research should focus on the geographical generalisability of these findings, and the applicability for PR programmes for other CRDs.

**Author affiliations**
[1]Respiratory medicine, National Center for Cardiology and Therapy named after academician Mirsaid Mirrakhimov under the Ministry of Health of the Kyrgyz Republic, Bishkek, Kyrgyzstan
[2]Respiratory Medicine, Intensive Care and Sleep Medicine Department, Republican Research Center of Pulmonology and Rehabilitation, Bishkek, Kyrgyzstan
[3]Department of Respiratory Sciences, University of Leicester Department of Respiratory Sciences, Leicester, UK
[4]Centre for Exercise and Rehabilitation Science, NIHR Leicester Biomedical Research Centre – Respiratory, University Hospitals of Leicester NHS Trust, University Hospitals of Leicester NHS Trust, Leicester, UK
[5]Faculty of Health, University of Plymouth, Plymouth, UK
[6]School of Sport, Exercise and Health Sciences, Loughborough University, Loughborough, UK

**Contributors** MM: conducting interviews (data collection), thematic analysis, data interpretation and writing an article. GM: conducting interviews (data collection), supporting thematic analysis, data interpretation and writing an article. ZKY: supporting thematic analysis, data interpretation and writing an article. MO: supporting thematic analysis, data interpretation and writing an article. CB: supporting thematic analysis and writing an article. AA: supporting thematic analysis and writing an article. AVJ: assisting with interview scheduling and data interpretation. RBM: assisting with interview scheduling and data interpretation. RJ: supporting with data interpretation and data analysis. AB: supporting thematic analysis and writing an article. DM: supporting thematic analysis, data interpretation and writing an article. TS: development of the concept and design of the study and writing an article. SJS: development of the concept and design of the study and writing an article

**Funding** This research was funded by the National Institute for Health Research (NIHR) (17/63/20) using UK aid from the UK Government to support global health research.

**Competing interests** None declared.

**Patient consent for publication** Not applicable.

**Ethics approval** The study was approved 22/07/2019 by Ethics Committee NCCIM (ref: no. 17) and by University of Leicester ethics committee (ref: no. 22293).

**Provenance and peer review** Not commissioned; externally peer reviewed.

**Data availability statement** Data are available upon reasonable request. There is a protocol paper that contains a protocol for all research work packages, including qualitative study. This protocol paper has been submitted to the BMJ Open and is currently awaiting publication.

**ORCID iDs**
Maamed Mademilov http://orcid.org/0000-0001-8528-3115
Zainab K Yusuf http://orcid.org/0000-0001-7859-5102
Mark W Orme http://orcid.org/0000-0003-4678-6574
Azamat Akylbekov http://orcid.org/0000-0001-5761-399X
Amy V Jones http://orcid.org/0000-0001-6565-8645

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
