## [Reviewer comments · BMJ Open]

ARTICLE DETAILS

TITLE (PROVISIONAL)	What should pulmonary rehabilitation look like for people living with post-tuberculosis lung disease in the Bishkek and Chui region of the Kyrgyz Republic?: A qualitative exploration
AUTHORS	Mademilov, Maamed; Mirzalieva, Gulzada; Yusuf, Zainab; Orme, Mark; Bourne, Claire; Akylbekov, Azamat; Jones, Amy; Miah, Ruhme; Jones, Rupert; Barton, Andy; Malcolm, Dominic; Sooronbaev, Talant; Singh, Sally

VERSION 1 – REVIEW

REVIEWER	Nolan, Claire Royal Brompton & Harefield, Department of Respiratory Medicine
REVIEW RETURNED	23-Jun-2021

GENERAL COMMENTS	Mademilov and colleagues have written a manuscript on qualitative research undertaken to understand the perceptions and opinions of healthcare professionals that will refer to pulmonary rehabilitation and people living with post-TB lung disease regarding setting up and delivering pulmonary rehabilitation in Bishkek and the Chuy region of Kyrgyzstan. The authors identify a number of factors that must be considered to ensure the culturally appropriate adaption of pulmonary rehabilitation for this region specifically relating to exercise and education. This is a well-written and interesting manuscript, I suggest it should be accepted for publication. Major comments: 1. Although post-Tb lung disease is clearly a significant problem in Kyrgyzstan, why were other lung conditions excluded from this research? It is my understand that smoking is prevalent in Kyrgyzstan and accordingly there is a significant proportion of people living with COPD. Are the authors of this manuscript doing a dis-service to people with other chronic lung diseases and contributing to further health inequality by not including them in this research? In other countries, pulmonary rehabilitation was originally developed for people with COPD and one of the consequences of this practice is that some people with other lung diseases believe that pulmonary rehabilitation does not address their needs. It is important that Kyrgyz researchers, clinicians and patients have ownership of this project, but also important to learn from the mistakes of other countries.2. The section on PPI involvement would benefit from including how people with post-TB lung disease were involved in the research question, study design and dissemination.
--

	3. Were equal numbers of people involved in the focus groups recruited from the different hospitals, which may ensure they are representative of the different groups of the Chuy region e.g. different socio-economic status etc? 4. It would be helpful if more information was provided on how the interviews and focus groups were conducted. As it currently reads, it doesn't sound like there were too many differences? 5. When the transcripts were translated into English, how was accuracy, specifically regarding Kyrgyz expression (language) and culture, assessed? 6. In the results section one person briefly says "many migrate" when talking about uptake of pulmonary rehabilitation. I'm assuming this refers to nomadic culture. This wasn't incorporated into the conclusions of the research and I wonder if the authors are missing an important aspect of Kyrgyz culture? I had expected to read more about the specifics of how pulmonary rehabilitation could be (re)started in Kyrgyzstan including consideration of specific challenges to the country e.g. migrant populations and those living in remote areas. Perhaps these issues are not relevant to the Chuy region or the population studied was not representative. Either way, it would be interesting to consider these points in more details and amend the title of the manuscript from "Kyrgyzstan" to "Bishkek and the Chuy region, Kyrgyzstan". 7. The authors cite two limitations. 1) Small sample size – I disagree; there was a large number of participants in the focus groups and interviews. In my understanding, 10 to 12 participants usually provide sufficient information to achieve data saturation. However, if the authors feel that data saturation was not achieved, this should be reported. 2) Greater proportion of female healthcare professionals – I disagree that this is a limitation because, as the authors report, this is a predominantly female specialty. Accordingly, the sample is likely to be representative. Minor comments: 8. The third and fourth paragraphs of the introduction are very interesting, and I suggest adding references to these sections. 9. The first paragraph of the results section could introduce the themes identified from the qualitative research, to ensure "flow" of this section.
--	--

REVIEWER	Cheng, Sonia The University of Sydney
REVIEW RETURNED	06-Sep-2021

GENERAL COMMENTS	This qualitative study used semi-structured interviews and focus groups to explore the design and content of culturally appropriate pulmonary rehabilitation (PR) programs in Kyrgyzstan, and potential barriers to implementation, with healthcare professionals who may refer to pulmonary rehabilitation (PR) and with people with post tuberculosis lung disease (PTBLD). Overall, the study is well-written with sound methodology. The study provides rich and valuable insight into critical intervention components and delivery, including preferred modes of exercise training, and potential solutions to increase adherence to PR in this clinical population. A more detailed description of the participants' characteristics is required to apply the findings of this study to other
---

	geographical/cultural or low-resource contexts. A diagram or table would also be useful to compare/contrast views of the healthcare professionals with people with PTBLD for the main themes. I have a few suggestions to improve the manuscript. Abstract: Major comments:  1. Please provide a brief description of the study participants (age, sex, type of healthcare professional, lung function if available). 2. Page 4, Line 27: “The culture-specific and population-specific issues demonstrate the need to...potentially expanding PR components beyond international guidelines”: I feel this is overstating the study’s conclusions since (i) this study did not specifically evaluate current models of PR delivered in Western countries and whether or not they would be culturally appropriate in Kyrgyzstan, and (ii) the themes in this study did not contradict currently known attitudes/barriers to PR. It may be more appropriate to state that addressing stigma and providing certain types of exercise training/education modules is important for this specific clinical population. Introduction: This section is well written and provides an excellent overview of the historical and current issues faced by people with PTBLD. I only have minor comments for this section. Minor comments:  1. Could the authors please clarify: Are there no existing PR programs in this geographical region OR is PR is not readily available for the clinical population? If the latter, it would be helpful to understand what structure and delivery of PR is currently available, even if for other lung diseases. 2. Page 7, Line 6: “This study combines the views of potential PR referrers and adults living with PTBLD in Kyrgyzstan to identify both the potential demand...for PR”: ‘Acceptability’ may be a more appropriate term than ‘demand’ here, as the need for PR in this population is well illustrated in earlier paragraphs. Methodology: Major comments:  1. Page 7, Line 53: Please provide more detail on how participants with PTBLD were recruited to the study and what sampling strategy was used. 2. Page 8, Line 18: Please provide a copy of the interview guide. Was the same interview guide used for both healthcare professionals and people with PTBLD? 3. Data analysis: Please comment on data saturation: Was data saturation reached? Was a specific number of participants determined a priori to reach saturation? Minor comments:  1. Page 7, Line 43: Please specify the sampling strategy used in this study. It appears to be a convenience sample. What type of healthcare professionals were approached to participate in the study? Only pulmonologists and TB specialists? 2. Page 8, Line 12: Did the interviewer MM also act as the facilitator during the focus groups? 3. Page 8, Line 55: Were computer software packages such as NVivo used to create codes? Results:
--	--

	Major comments:  1. Before reporting on the main themes, the participants' characteristics need to be defined. Please provide a summary of demographic information to better understand the study sample. For example, for referrers: years and area of practice; previous referral to rehabilitation. For participants: Lung function, symptom score, SES, previous attendance at rehabilitation. Minor comments:  2. The comparison of data between healthcare professionals and people with PTBLD is well written and interwoven throughout the text; however, a diagram or table summarising the views of each group would make it easy for readers to compare/contrast the data. Discussion: This section is well written and highlights important considerations for the design and content of future PR programs in Kyrgyzstan. I only have minor comments for this section. Minor comments:  1. Page 16, Line 26: The paragraph describing the available treatment for PTBLD is better suited to the introduction. 2. Page 17, Line 17: It is interesting that participants preferred modes of exercise training that are do not require expensive equipment and could be undertaken in home or community settings, such as walking, dance and balls sports. This supports the feasibility of PR programs in this local setting and should be mentioned in the discussion. 3. Page 18, Line 42: "Unlike dancing, singing was equivocally received by research participants": It is unclear what equivocally received means – consider rewording for clarity. It seems that singing may be beneficial intervention to include as part of PR. See the Cochrane review by McNamara (2017) on benefits of singing on HRQOL and dyspnoea (https://doi.org/10.1002/14651858.CD012296.pub2). 4. Page 19, Line 3: Consider moving this final paragraph from the limitations section to the beginning of the discussion. This paragraph highlights the novelty of this study and its clinical implications for establishing PR in other low-resource contexts. Conclusion: Major comments:  1. Page 19, Line 26: "Consequently, new PR services can go beyond international guidelines based almost exclusively on Western norms to ensure services are attractive to local populations whilst keeping exercise and education at its core." As stated previously for the abstract, I believe the study is overstating its conclusions with the statement "going beyond international guidelines based almost exclusively on Western norms". It would be more appropriate to highlight the key needs for this population, ie, (i) addressing stigma through psychosocial support, and (ii) in this specific context low-resource programs with culturally appropriate education/exercise modules are more likely to be successful. 2. The conclusion should state directions for future research, eg, using this study to provide a framework for real-world implementation of PR programs and evaluating the effectiveness of such programs for people with PTBLD.
--	---

VERSION 1 – AUTHOR RESPONSE

Reviewer: 1

Dr. Claire Nolan, Royal Brompton & Harefield

Major comments:

1. Although post-Tb lung disease is clearly a significant problem in Kyrgyzstan, why were other lung conditions excluded from this research? It is my understanding that smoking is prevalent in Kyrgyzstan and accordingly there is a significant proportion of people living with COPD. Are the authors of this manuscript doing a dis-service to people with other chronic lung diseases and contributing to further health inequality by not including them in this research? In other countries, pulmonary rehabilitation was originally developed for people with COPD and one of the consequences of this practice is that some people with other lung diseases believe that pulmonary rehabilitation does not address their needs. It is important that Kyrgyz researchers, clinicians and patients have ownership of this project, but also important to learn from the mistakes of other countries.

There are indeed many patients with COPD and other chronic respiratory diseases in the Kyrgyz Republic, but it is important to note that ordinary patients with such diseases can easily access medical services and other activities. For patients with tuberculosis, there are separate hospitals, and after recovery they, due to stigmatization from society and medical workers, including, have difficulties in obtaining medical services, since when they go to ordinary polyclinics, they require a certificate of full recovery from tuberculosis. Also, our pulmonary rehabilitation program includes adapted features for patients with post-tuberculosis lung diseases, such as tuberculosis education, psychological support. Also, the results of this experience can be implemented on the territory of post-Soviet countries, where there is a similarity in the health care system.

But the reviewer is right, it is impossible to separate patients for pulmonary rehabilitation, we must take this into account, and we believe that it is necessary to introduce a pulmonary rehabilitation program in the health care system as a whole for patients with chronic respiratory diseases, because this is not given sufficient attention in our country, compared with drug treatment.

The key points here are that it is the most challenging lung condition to address, because of public fears about contagion which lead to stigmatization which is evidenced by the issuing of a certificate of full recovery for these patients.

2. The section on PPI involvement would benefit from including how people with post-TB lung disease were involved in the research question, study design and dissemination.

Patients were not directly involved in formulating the interview questions or research question in the traditional sense of patient and public involvement. It was through informal conversations with patients and clinical consultations with patients that emphasized the importance of directly involving patients in the design of their PR; not least because of the stigma and alienation experienced as a result of their PTBLD. Patients who were involved in the research and resulting PR trial will be invited to support dissemination activities. By the nature of our qualitative research, the findings of the present study are being used to develop a patient-informed PR programme in accordance with the principles of PPI.

3. Were equal numbers of people involved in the focus groups recruited from the different hospitals, which may ensure they are representative of the different groups of the Chuy region e.g. different socio-economic status etc?

In the course of the selection and involvement of referrers in this process, participants were included in this study from all centers, both in the city of Bishkek and in Chui oblast (region). Another important point is that such patients receive basic treatment in tuberculosis hospitals, where the same conditions are created for all of them, which unites them all. Also, the participants had different socio-economic status, some of them lived in urban conditions, some came from remote villages, there were homeless people among them and many others, but the monitoring of an equal number from different centers was not carried out. Based on the above, the participants can be considered broadly

representative of the range of patient background throughout the Kyrgyz Republic, as well as for most post-Soviet countries.

We probably need to be more specific here – that the study is representative of the Chuy region of Kyrgyzstan and while we did not purposively sample according demographic characteristics, the study sample included people from a range of different backgrounds (e.g. those from urban and rural locations, homeless as well as employed).

Additionally, we should add this to the section on limitation – that the study is representative of the Chuy region of Kyrgyzstan and not Kyrgyzstan as a whole.

4. It would be helpful if more information was provided on how the interviews and focus groups were conducted. As it currently reads, it doesn't sound like there were too many differences?

Interviews with health workers were conducted at their workplace in various health care facilities, both in urban (national center, city center) and rural settings, face to face. Focus group discussions: participants from different points were invited to the NCCIM, where the interview was conducted, and during the interview, participants could freely interact with each other, ask each other questions and answer them.

I think the point to stress here was that the FGs were more open ended and involved interaction between participants and consequently they lasted longer. Can we add an example of how the participants interacted? Can we explain that particular themes emerged through the FG discussions that we had not fully anticipated – like the issues around stigma

And the need for peer-to-peer support? I think I remember how patients actually benefited from/enjoyed the FGDs themselves... Dom?5. When the transcripts were translated into English, how was accuracy, specifically regarding Kyrgyz expression (language) and culture, assessed?

The translation was carried out by a Kyrgyz team that understood and interpreted all expressions in an accessible and understandable form, and also conveyed such details as laughter, expressions, etc. Special phraseological units were converted into an accessible form for understanding in English.

6. In the results section one person briefly says “many migrate” when talking about uptake of pulmonary rehabilitation. I'm assuming this refers to nomadic culture. This wasn't incorporated into the conclusions of the research and I wonder if the authors are missing an important aspect of Kyrgyz culture? I had expected to read more about the specifics of how pulmonary rehabilitation could be (re)started in Kyrgyzstan including consideration of specific challenges to the country e.g. migrant populations and those living in remote areas. Perhaps these issues are not relevant to the Chuy region or the population studied was not representative. Either way, it would be interesting to consider these points in more details and amend the title of the manuscript from “Kyrgyzstan” to “Bishkek and the Chuy region, Kyrgyzstan”.

Unfortunately, at this point in time, the nomadic culture does not have a strong influence, only not numerous residents use it in reality on pastures for animal husbandry. In this context, it is said about the migration of our citizens to other countries to earn money, this does not imply a nomadic lifestyle, many of our citizens, due to financial difficulties, are forced to leave the country for an indefinite period, but they can return in case of illness, to receive medical care, just such a case was described by the referrer. The socio-economic status of all regions is approximately at the same level, with the exception of large cities. So in our case, among the participants were urban residents, as well as residents of remote rural areas, including those staying at work in the city from other regions. The experience gained can be disseminated both in the Chui region and the city of Bishkek, and in all other regions, there is no big difference, the studied group was representative.

‘It was noted that many migrate (primarily for economic reasons) after finishing treatment.’

7. The authors cite two limitations. 1) Small sample size – I disagree; there was a large number of participants in the focus groups and interviews. In my understanding, 10 to 12 participants usually provide sufficient information to achieve data saturation. However, if the authors feel that data saturation was not achieved, this should be reported. 2) Greater proportion of female healthcare professionals – I disagree that this is a limitation because, as the authors report, this is a predominantly female specialty. Accordingly, the sample is likely to be representative.

We remove these as limitations, and then add new ones – Following on from what this reviewer has said we could list as limitations – that the study is not representative of the Kyrgyzstan population as a whole, and it is focused on PR for PTBLD rather than a broader range of lung conditions

8. The third and fourth paragraphs of the introduction are very interesting, and I suggest adding references to these sections.

This data was collected through the interactions with medical professionals who worked at these centers at the time, unfortunately, we could not find any supporting references to this data.

9. The first paragraph of the results section could introduce the themes identified from the qualitative research, to ensure “flow” of this section.

Referrers and respondents living with PTBLD were positive about the potential introduction of PR. They identified a range of barriers/facilitators that would need to be considered in the design and implementation of any such program. A range of potential PR components were identified, as well as the need to tailor PR to individual physical conditions, provide effective communication and education, and be sensitive to the patients’ psychosocial state. It was also noted the importance of including psychological support in the PR, which will be specific to patients with post-TB disease who have stigma and psychological problems. Respondents living with PTBLD shared their illness experiences, such as symptom burden, challenges faced, and patterns of exercise and physical activity, and provided a vision of how a PR program should best be designed.

Reviewer: 2

Dr. Sonia Cheng, The University of Sydney, Macquarie University Hospital

Abstract:

Major comments:

1. Please provide a brief description of the study participants (age, sex, type of healthcare professional, lung function if available).

2. Page 4, Line 27: “The culture-specific and population-specific issues demonstrate the need to...potentially expanding PR components beyond international guidelines”: I feel this is overstating the study’s conclusions since (i) this study did not specifically evaluate current models of PR delivered in Western countries and whether or not they would be culturally appropriate in Kyrgyzstan, and (ii) the themes in this study did not contradict currently known attitudes/barriers to PR. It may be more appropriate to state that addressing stigma and providing certain types of exercise training/education modules is important for this specific clinical population.

1. We have limited information on this. Word limit of the abstract restricts us from adding more detail

2. Original text: «The culture-specific and population-specific issues highlighted in this work demonstrate the need to adapt services to the local context, potentially expanding PR components beyond international guidelines to make services attractive to those they seek to help and those involved in the delivery. »

New text: ‘The culture-specific and population-specific issues highlighted in this work demonstrate the need to address stigma and provide certain types of exercise training/education modules for this specific clinical population. In other respects the currently known attitudes/barriers to PR, identified in Western research, appear to apply’

Introduction:

This section is well written and provides an excellent overview of the historical and current issues faced by people with PTBLD. I only have minor comments for this section.

Minor comments:

1. Could the authors please clarify: Are there no existing PR programs in this geographical region OR is PR is not readily available for the clinical population? If the latter, it would be helpful to understand what structure and delivery of PR is currently available, even if for other lung diseases.

2. Page 7, Line 6: “This study combines the views of potential PR referrers and adults living with PTBLD in Kyrgyzstan to identify both the potential demand...for PR”: ‘Acceptability’ may be a more appropriate term than ‘demand’ here, as the need for PR in this population is well illustrated in earlier paragraphs.

1. Unfortunately, pulmonary rehabilitation is not implemented anywhere in Central Asia.

2. Original text: «Consequently, this study combines the views of potential PR referrers and adults living with PTBLD in Kyrgyzstan to identify both the potential demand and most appropriate design for PR»

New text: «Consequently, this study combines the views of potential PR referrers and adults living with PTBLD in Kyrgyzstan to identify both the potential acceptability and most appropriate design for PR»

Methodology:

Major comments:

1. Page 7, Line 53: Please provide more detail on how participants with PTBLD were recruited to the study and what sampling strategy was used.

2. Page 8, Line 18: Please provide a copy of the interview guide. Was the same interview guide used for both healthcare professionals and people with PTBLD?

3. Data analysis: Please comment on data saturation: Was data saturation reached? Was a specific number of participants determined a priori to reach saturation?

Minor comments:

1. Page 7, Line 43: Please specify the sampling strategy used in this study. It appears to be a convenience sample. What type of healthcare professionals were approached to participate in the study? Only pulmonologists and TB specialists?

2. Page 8, Line 12: Did the interviewer MM also act as the facilitator during the focus groups?

3. Page 8, Line 55: Were computer software packages such as NVivo used to create codes?

Major comments:

1. Sampling strategy: in our study we used a combination of targeted and convenience sampling. We assumed that the participants had to be selected in medical centers (tuberculosis hospitals), according to the inclusion criteria: they must have confirmation of completed treatment, etc..

2. Different interview guides were used for interviews and focus groups, attached.

3. ‘Recruitment targets were determined a priori according to the norms of the respective methods. Partly due to the delays imposed by transcription, translation and cross-national analysis, but also due to the relative ease of access, we recruited up to these targets. Data saturation was achieved, but known only retrospectively as the data analysis progressed.’

Minor comments:

1. In our study, we used a combination of targeted and convenient sampling.

When sampling, we were guided by the research protocol, according to which, among medical specialists, we selected only those who directly in their clinical practice deal with PTBLD patients, these are TB specialist and pulmonologists.

2. Interviewer MM acted as facilitator during focus groups

3. Such programs were not used

Results:

Major comments:

1. Before reporting on the main themes, the participants’ characteristics need to be defined. Please provide a summary of demographic information to better understand the study sample. For example, for referrers: years and area of practice; previous referral to rehabilitation. For participants: Lung function, symptom score, SES, previous attendance at rehabilitation.

Minor comments:

2. The comparison of data between healthcare professionals and people with PTBLD is well written and interwoven throughout the text; however, a diagram or table summarising the views of each group would make it easy for readers to compare/contrast the data.

1. Demographic information, for referrers, we can only provide information on the field of clinical practice, since we included in the study only medical specialists working with post-TB patients, pulmonologists and TB specialists, we can also say with confidence that they have no previous experience in referring such patients to rehabilitation. Regarding focus group participants, we cannot provide information on age, lung function, and none of the participants has any previous experience of attending PR

2. We will include table

Discussion:

This section is well written and highlights important considerations for the design and content of future PR programs in Kyrgyzstan. I only have minor comments for this section.

Minor comments:

1. Page 16, Line 26: The paragraph describing the available treatment for PTBLD is better suited to the introduction.

2. Page 17, Line 17: It is interesting that participants preferred modes of exercise training that are do not require expensive equipment and could be undertaken in home or community settings, such as walking, dance and balls sports. This supports the feasibility of PR programs in this local setting and should be mentioned in the discussion.

3. Page 18, Line 42: "Unlike dancing, singing was equivocally received by research participants": It is unclear what equivocally received means – consider rewording for clarity. It seems that singing may be beneficial intervention to include as part of PR. See the Cochrane review by McNamara (2017) on benefits of singing on HRQOL and dyspnoea (<https://doi.org/10.1002/14651858.CD012296.pub2>).

4. Page 19, Line 3: Consider moving this final paragraph from the limitations section to the beginning of the discussion. This paragraph highlights the novelty of this study and its clinical implications for establishing PR in other low-resource contexts.

1. At present, the treatment process for these individuals is exclusively pharmacological (lasting between 6 months and 2 years), and with no post-treatment care pathway, they are often left in a state of limbo whereby they are no longer infectious but are living with chronic symptoms and associated stigma. We will move to introduction.

2. It is interesting that participants preferred modes of exercise training that are do not require expensive equipment and could be undertaken in home or community settings, such as walking, dance and balls sports. We will move to discussion.

3. 'Unlike dancing, singing had a mixed reception from participants some of whom were enthusiastic and some of whom said it would not be popular.'

4. Kyrgyzstan is leading the development of PR among many other post-Soviet countries and in Central Asia, where there is currently an absence of rehabilitation services, and therefore ideas about how to successfully implement PR may have significant implications for other LMIC or low-resource contexts. We will move to the beginning of the discussion.

Conclusion:

Major comments:

1. Page 19, Line 26: "Consequently, new PR services can go beyond international guidelines based almost exclusively on Western norms to ensure services are attractive to local populations whilst keeping exercise and education at its core." As stated previously for the abstract, I believe the study is overstating its conclusions with the statement "going beyond international guidelines based almost exclusively on Western norms". It would be more appropriate to highlight the key needs for this population, ie, (i) addressing stigma through psychosocial support, and (ii) in this specific context low-resource programs with culturally appropriate education/exercise modules are more likely to be successful.

2. The conclusion should state directions for future research, eg, using this study to provide a framework for real-world implementation of PR programs and evaluating the effectiveness of such programs for people with PTBLD.

1. «Addressing stigma through psychosocial support, and in this particular context, low-resource programs with culturally appropriate education / exercise modules are more likely to be successful» See my response to on the abstract. I think they are right and so we just need to change the abstract and conclusion

2. The results of this study can be applied to provide a basis for the real implementation of PR programs and to evaluate the effectiveness of such programs for people with PTBLD.

VERSION 2 – REVIEW

REVIEWER	Nolan, Claire Royal Brompton & Harefield, Department of Respiratory Medicine
REVIEW RETURNED	01-Dec-2021

GENERAL COMMENTS	The authors have addressed my questions. Congratulations on an interesting piece of research and I look forward to reading about the implementation of this programme.
--

REVIEWER	Cheng, Sonia The University of Sydney
REVIEW RETURNED	13-Dec-2021

GENERAL COMMENTS	I congratulate the authors on their improved manuscript. It should be acknowledged in the study limitations that there is a lack of data on participants' characteristics to understand their demographics and severity of disease, and thus unclear whether the study findings can be generalised to other people with PTBLD in Kyrgyzstan. I understand this lack of data to define participant characteristics is attributed to the absence of holistic treatment for this patient group and the challenges of providing care. The authors have addressed all my remaining comments.
---

VERSION 2 – AUTHOR RESPONSE

Reviewer: 1

Dr. Claire Nolan, Royal Brompton & Harefield

Comments to the Author:

The authors have addressed my questions. Congratulations on an interesting piece of research and I look forward to reading about the implementation of this programme.

Reviewer: 2

Dr. Sonia Cheng, The University of Sydney, Macquarie University Hospital

Comments to the Author:

I congratulate the authors on their improved manuscript.

It should be acknowledged in the study limitations that there is a lack of data on participants' characteristics to understand their demographics and severity of disease, and thus unclear whether the study findings can be generalised to other people with PTBLD in Kyrgyzstan. I understand this lack of data to define participant characteristics is attributed to the absence of holistic treatment for this patient group and the challenges of providing care.

The authors have addressed all my remaining comments.

Reviewer: 1

Competing interests of Reviewer: Nil

Reviewer: 2

Competing interests of Reviewer: I have no competing interests.

Reply to Reviewers

Strengths and limitations of this study

- The opinions of patients with PTBLD and HCPs (TB specialists and pulmonologists) were deliberately taken into account to develop a pulmonary rehabilitation program.
- Participants were diverse in terms of demographic characteristics, severity of the disease, experience and social status.
- The study was conducted in one geographical region in Kyrgyzstan, therefore the implications reported need to be tested to show applicability to other cultural contexts.
- The study focused on PR for PTBLD rather than the broader spectrum of lung diseases.

Limitations

The study was conducted in one geographical region in Kyrgyzstan, therefore the applicability of the implications reported require empirical verification. The absence of holistic treatment for this group of patients and the problems of care do not provide a complete picture to understand their demographic characteristics and severity of the disease, and thus it is unclear if the study results can be generalized to other people with PTBLD in Kyrgyzstan. The study focused on PR for PTBLD rather than the broader spectrum of lung diseases, but it is important to note that there is no evidence to suggest that the implementation of PR for PTBLD is likely to be easier in relation to this group, due to the stigma associated with the condition.